# Change in Tissue Microbiome and Related Human Beta Defensin Levels Induced by Antibiotic Use in Bladder Carcinoma

**DOI:** 10.3390/ijms25084562

**Published:** 2024-04-22

**Authors:** Ádám Monyók, Bassel Mansour, István Vadnay, Nóra Makra, Zsuzsanna A. Dunai, Éva Nemes-Nikodém, Balázs Stercz, Dóra Szabó, Eszter Ostorházi

**Affiliations:** 1Department of Urology, Markhot Ferenc University Teaching Hospital, 3300 Eger, Hungary; adammonyok@gmail.com (Á.M.); bassel.mansour0@gmail.com (B.M.); 2Department of Pathology, Markhot Ferenc University Teaching Hospital, 3300 Eger, Hungary; vadnay.istvan@gmail.com; 3Department of Medical Microbiology, Semmelweis University, 1085 Budapest, Hungary; makra.nora@med.semmelweis-univ.hu (N.M.); zsuzsanna.dunai@gmail.com (Z.A.D.); nemes-nikodem.eva@med.semmelweis-univ.hu (É.N.-N.); stercz.balazs@med.semmelweis-univ.hu (B.S.); szabo.dora@med.semmelweis-univ.hu (D.S.); 4Neurosurgery and Neurointervention Clinic, Semmelweis University, 1085 Budapest, Hungary; 5Department of Dermatology, Venereology and Dermatooncology, Semmelweis University, 1085 Budapest, Hungary

**Keywords:** antibiotic treatment, bladder cancer, human beta defensins, tissue microbiome

## Abstract

It is now generally accepted that the success of antitumor therapy can be impaired by concurrent antibiotic therapy, the presence of certain bacteria, and elevated defensin levels around the tumor tissue. The aim of our current investigation was to identify the underlying changes in microbiome and defensin levels in the tumor tissue induced by different antibiotics, as well as the duration of this modification. The microbiome of the tumor tissues was significantly different from that of healthy volunteers. Comparing only the tumor samples, no significant difference was confirmed between the untreated group and the group treated with antibiotics more than 3 months earlier. However, antibiotic treatment within 3 months of analysis resulted in a significantly modified microbiome composition. Irrespective of whether Fosfomycin, Fluoroquinolone or Beta-lactam treatment was used, the abundance of *Bacteroides* decreased, and *Staphylococcus* abundance increased. Large amounts of the genus *Acinetobacter* were observed in the Fluoroquinolone-treated group. Regardless of the antibiotic treatment, hBD1 expression of the tumor cells consistently doubled. The increase in hBD2 and hBD3 expression was the highest in the Beta-lactam treated group. Apparently, antibiotic treatment within 3 months of sample analysis induced microbiome changes and defensin expression levels, depending on the identity of the applied antibiotic.

## 1. Introduction

The relationship between the formation, progression, detection markers, treatment options, and the composition of the microbiome of bladder carcinoma is a frequently investigated research topic [1,2,3]. To characterize the underlying phenomena, spontaneously voided urine is a patient-friendly sampling option that can be repeated frequently without surgical intervention. Recently, we documented that the microbiome detected in the urine is not a mirror image of the microbiome found in the tumor tissue [4]. In addition, the tissue microbiome is specific to a particular patient, and its composition is constant at different points of the tumor [4]. We also found significant differences in the microbiome composition of healthy and cancerous tissues [5]. These findings were in line with numerous reports from other laboratories [6,7,8]. Surgery-originated tissue samples cannot be replaced for accurate analysis of the relationship between the tumor and the microbiome.

Antibiotics play a dual role in cancer therapy, acting as both beneficial and detrimental agents [9]. While Floxacin prevents chemoresistance and metastasis in lung and bladder carcinoma [10], some other antibiotics—like chloramphenicol, azithromycin, linezolid, or tigecycline—have anticancer effects inducing mitochondrial dysfunction in tumor stem cells [11]. Having said that, by disrupting the gut microbiota, antibiotics potentially reduce the body’s ability to fight cancer [12,13]. Antibiotics can cause a change in the gut microbiome, which affects the immune cells and impairs the efficacy of BCG installation or success of check-point inhibitor treatments [14,15]. It remains to be answered whether only changes in the intestinal microbiome influence the effectiveness of antitumor treatment, or the microbiome of the immediate environment of the bladder cancer also affects the therapy prognosis.

The formation of an individual’s microbiome is influenced by many factors, including age, gender, dietary habits, underlying diseases and medications. The composition of the gut and urine microbiome can be changed either by antibiotic or probiotic treatments [16,17]. Changes in the gut microbiome after fecal microbiota transplantation (FMT) can indirectly modify the urinary microbiome [18]. As far as we understand, no previous research addressed the question as to how the composition of the microbiome detectable in bladder tumor tissue can be influenced by antibiotic treatment. Although, due to understandable ethical and patient compliance reasons, it is not possible to follow the changes in the tissue microbiome of a given patient as a result of antibiotic treatment, we had the opportunity to compare histological samples of untreated and antibiotic-treated patients. By comparing the tissue microbiome of patients without antibiotic treatment and those who received antibiotic treatment within 3 months of sampling or more than 3 months, we retroactively analyze the existence and duration of the relationships between tissue microbiome composition and antibiotic treatment. The most commonly used antibiotics against urinary tract infections are Fosfomycin (FOS), Fluoroquinolones (FQ), and Beta-lactams (BL). We examined whether the type of antibiotic used affected the quality of microbiome change and the level of expressed defensins.

Human beta defensins (hBDs) are antibacterial peptides produced by various cells of the human body, which have different effects on the progression of tumors and the success of anticancer treatment. They play a role in host immunity and mucosal surface protection against fungal, bacterial, and viral pathogens [19]. While human beta defensin 1 is produced constitutively, expression of hBD2 and hBD3 is induced by bacteria [20]. The amounts of hBD produced and the components of the microbiome are completely interrelated. The autonomous defensin production of tumor cells may also amend this dynamic to an unpredictable degree and direction. There is also evidence that hBDs may have a role in cancer, as they can promote or inhibit cancer cell proliferation and migration depending upon the origin of the cancer cell [21]. For example, hBD1 appears to be downregulated in prostate and renal clear cell carcinomas but upregulated in lung carcinoma. Additionally, hBD1 has been suggested as a candidate tumor suppressor in oral, renal, and prostate cancers [22]. In cervical cancer, hBD2 and hBD3 exhibit elevated levels and increase the migration and proliferation of cancer cells [23]. Defensins can also act as chemoattractants for immune cells in the tumor microenvironment. Earlier, any relationship between hBDs and bladder cancer were found only for hBD1 and hBD2. Exogenous synthetic hBD1 peptide inhibits bladder cancer cell migration and proliferation [21], and urine-derived hBD1 has been shown to suppress bladder cancer cell growth [23,24]. Additionally, hBD1 has been studied for its potential clinical applications in bladder cancer [25]. For hBD2, the relationship with bladder cancer was described as elevated hBD2 levels inhibit the success of Bacillus Calmétte-Guérin (BCG) treatment [26,27].

In addition to changes in the microbiome, changes in the related hBD levels can also reduce the effectiveness of tumor therapy during antibiotic treatment Figure 1.

In the current study, we compared the microbiome associated with the bladder mucosa of 10 benign prostate hyperplasia (BPH) patients and the amounts of human beta defensin (hBD) 1-2-3 detected in patient tissues as negative controls. The microbiome and defensin expression values of bladder carcinoma patients (BC) were also compared. In addition, we investigated the presence or absence of antibiotic therapy and its timing, as well as how the identity of the antibiotics influence the obtained values. The 60 patients in the BC group were split into the following subgroups: 26 patients without antibiotic therapy (BC AB0), 26 patients with antibiotic therapy within 3 months (BC AB3), and 8 patients with antibiotic therapy more than 3 months before analysis (BC AB6).

Within the BC AB3 group, nine patients received Fosfomycin (FOS) treatment, seven patients received Fluoroquinolone (FQ) treatment, and ten patients received Beta-lactam (BL) treatment. The study design is shown in Figure 2.

Our current research aims to determine whether antibiotic therapy induces changes in the microbiome composition of tumor tissue.

Another goal is to determine how durable the antibiotic-induced microbiome change is over time.

We aim to identify changes in the microbiome in response to treatment with the most commonly used antibiotics.

We also aim to characterize the defensin level changes associated with changes in the microbiome.

This study can highlight the prognosis of tumor therapy driven by the tissue microbiome composition and defensin levels, and a correlation of these factors with possible changes induced by antibiotic treatment.

## 2. Results

### 2.1. Patient Characteristics

Sixty bladder cancer (BC) patients of the Department of Urology, Markhot Ferenc University Teaching Hospital, Eger, Hungary were enrolled between February and August of 2021. In 6 months before the surgical intervention, 26 patients remained uninfected, and thus did not receive prior antibiotic treatment (BC AB0). Another 34 patients received antibiotic therapy before the surgery, of which 26 patients received the therapy within 3 months of microbiome screening (BC AB3) and 8 patients received it more than 3 months earlier (BC AB6). Bladder mucosa samples from 10 patients with benign prostatic hyperplasia (BPH) were used as non-bladder carcinoma negative controls. The characteristics of the study participants are presented in Table 1.

### 2.2. Microbiome Composition in Healthy, Antibiotic-Untreated Bladder Cancer and Antibiotic-Treated Bladder Cancer Tissues

From 60 BC cancer tissue samples and 10 BPH mucosal samples, two smaller pieces were processed separately, for a total of *n* = 140 tested samples from 70 patients. A total of 47.9 million valid bacterial sequences were obtained, resulting in 36.7 million high-quality reads. The median number of reads in BC tissue samples was significantly higher (265,437 (IQR: 34,506)) (*p* = 0.001) than in BPH bladder mucosal samples (100,995 (IQR: 12,391)).

Although tumor tissue samples contained higher amounts of bacterial DNA than BPH mucosal samples, the alpha diversity of BPH samples was significantly higher than that of tumor samples. Figure 3A, the Chao1 alpha diversity diagram, shows that each BC group was significantly different from the BPH group, but there were no significant differences between individual BC groups.

The beta diversity of the BC groups is also significantly different from that of the BPH group. Bray–Curtis Principal Coordinate Analysis shows that among BC groups, antibiotic treatment within 3 months resulted in a significant beta diversity difference, but the group treated within 6 months no longer differed significantly from the untreated group (Figure 3B). Gender did not affect the diversity values associated with antibiotic treatment.

The abundance of *Proteobacteria* and *Actinobacteria* was significantly higher in the tumor groups than in the BPH group, regardless of the presence or timing of the antibiotic treatment. In contrast, the abundance of the *Bacteroidetes* phylum increased significantly in the BPH group (Figure 4A).

At the genus level, the median abundances of *Staphylococcus*, *Corynebacterium*, *Propionibacterium,* and *Acinetobacter* taxa were significantly higher in the cancer groups, regardless of the presence or timing of the antibiotic treatment. Conversely, *Bacteroides*, *Faecalibacterium*, *Blautia*, *Ruminococcus,* and *Parabacteroides* genera all had significantly higher abundances in the BPH group (Figure 4B).

The diversity results of the group treated with antibiotics within 3 months differed significantly from those of the untreated group and the group treated within 6 months. The significantly different diversity values of the AB3 group can be partially explained by the fact that single dominant genus types—with abundance values between 34 and 97%—were present in eight patients (Figure 5). The genera with exceptionally high abundance were *Ureaplasma* (95%), *Lactobacillus* (94%), *Staphylococcus* (97%), *Gardnerella* (59% or 92%), *Veillonella* (50%), *Enterococcus* (42%), and *Corynebacterium* (34%).

In the beta diversity diagram in Figure 3B, the AB3 group was separated from the AB0 and AB6 groups by another feature, that is the area that can be fitted to the AB3 values were located closer to the BPH group. The difference in the abundance of those genera that were significant between healthy and cancer samples, when examined the individual effects of antibiotics, did not differ significantly (Table A1).

Different antibiotic treatments did not result in a significant difference in alpha diversity values of the groups. By plotting the microbiome beta diversity of tissue samples from patients who received different antibiotic treatments within 3 months, apparently the microbiome change also depended upon the identity of the antibiotic. The beta diversity principal coordinate analysis based on the Bray–Curtis attribute (Figure 6) identified the microbiome of FQ and FOS treated subjects in significantly different groups compared to BL treated microbiome.

The relative abundance of the genera *Bacteroides*, *Faecalibacterium*, *Blautia*, *Parabacteroides*, and *Ruminococcus*, which characterize the healthy mucosal microbiome, decreased to varying degrees during antibiotic treatments. In all three groups treated with antibiotics, the *Staphylococcus* genus was present with a significantly higher relative abundance compared to the healthy microbiome. In the Fluoroquinolone-treated group, the 2nd most abundant taxon was *Acinetobacter*, which was absent in healthy mucosa. In the Fosfomycin-treated group, the genus with the second highest abundance was *Blautia*, which was also characteristic of the healthy mucosa, albeit the difference between the two groups was not significant. In the Beta-lactam-treated group, the second most common genus was *Bacteroides*, but the value of abundance was significantly lower compared to that of the healthy group. There was no significant difference between the bacterial abundance of the BC AB0 and BC AB6 groups in terms of any genera, but they differed significantly from the group treated with antibiotics within 3 months with a higher abundance of *Corynebacterium* (Figure 7A).

### 2.3. Human Beta Defensin Expression and the Relationship with Microbiome Composition

Considering the human beta defensin expression median value of the BPH tissue samples baseline, we examined the hBD expression of the tumor tissue samples depending upon pre-existing antibiotic treatment. Regardless of antibiotic treatment, median hBD1 expression was significantly increased in all tumor tissues, 2.62-fold in BC AB0 (*p* = 0.014), 2.54-fold in the BC AB6 (*p* = 0.041), 2.57-fold in the FQ (*p* = 0.047), 2.62-fold (*p* = 0.014) in the FOS, and 2.59-fold (*p* = 0.015) in the BL group. There was no significant difference between the values of antibiotic-treated and untreated groups. The median hBD2 expression increased 4.14-fold (*p* = 0.0005) in BC AB0 group, 3.98-fold (*p* = 0.004) in BC AB6, 1.85-fold (*p* = 0.02) in FQ group, 3.35-fold (*p* = 0.0004) in FOS, and 7.58-fold (*p* = 0.0006) in BL group. The biggest difference between the groups was in the increase in hBD3 levels; 7.32-fold (*p* = 0.004) in BC AB0 group, 7.63-fold (*p* = 0.004) in BC AB6, and 1.15-fold *(p* = 0.3), 1.71-fold (*p* = 0.03), 17.38-fold (*p* = 0.001) in FQ, FOS, and BL groups, respectively (Figure 7B).

## 3. Discussion

Even though urine sampling is a patient-friendly process and can be repeated often, it is unfortunately not suitable for examining the microbes affecting the tumor, because it does not have the same microbial composition as the tumor tissue itself. Previous studies have shown that it is preferable to use histological samples instead of urine as test material to investigate the relationship between bladder carcinoma and the microbiome [4,7]. In general, significant differences were detected between the microbiome of cancerous and healthy bladder mucosa samples [5,7].

During the 16S rRNA analysis of tumor and healthy tissue samples, we were able to detect a significantly higher amount of bacterial DNA with high-quality reads from the tumor tissue than from the healthy mucosa samples. This higher amount of bacterial DNA detectable in the tumor tissue was not only observed in the samples of patients without antibiotic therapy, but also in the samples of patients treated with antibiotics. It is possible that the bacteria attached to the cancerous mucosa were able to multiply because of the nutrients and microenvironment provided by the necrotic tissue debris, as well as the altered urine flow. It remains to be seen whether the amount of DNA detected comes from viable bacteria in the given location or from residual materials of dead bacteria adhered to the tumor mucosa. Substances released from living or dead bacteria can affect both tumor progression and the quality of the local immune response, including the expression of antibacterial peptides. Human beta defensins, as antibacterial peptides, have a critical role as gatekeepers of the pervasive and complex prokaryotic ecosystems that occupy all body surfaces and cavities, including the bladder mucosal surface [28].

Even though a higher amount of bacterial DNA was detected in the tumor tissue, the richness of bacterial taxa in the tumor was narrowed. Loss of microbiota alpha diversity has recently been described not only in colorectal tumors but also in bladder cancer [5,7,29]. The alpha diversity in all tumor samples was significantly lower than in healthy tissue. Certainly, this parameter of dysbiosis does not seem to depend on the presence and timing of antibiotic treatment.

Bray–Curtis beta diversity PCoA analysis revealed a significant differentiation in bacterial community composition among bladder cancerous and non-cancerous tissues. Furthermore, evaluating only the tumor samples, the beta diversity of the samples treated with antibiotics within three months also differed significantly from that of the rest of the samples. The tissue microbiome beta diversity of cancer patients receiving antibiotic treatment more than 3 months before the study no longer showed a significant difference from the tumor tissue value of the antibiotic-free group. Apparently, the individual microbiome changes caused by the presence of the tumor are only temporarily changed by bacterial infections and antibiotic treatment interventions, but after the infection and antibiotic effects have passed, the microbiome composition characteristic of the individual is re-established.

Although previous studies documented a significant difference in the composition of the bladder tumor tissue microbiome according to gender [4,5], the weakness of our present study is that we could only examine male bladder tissue samples as negative controls (BPH). Neither the microbiome change induced by the antibiotic treatment, nor the related defensin level changes were influenced by gender in this study. All female participants were already over 65 years old in the postmenopausal period, which may explain why the difference between the sexes was no longer detectable in the current comparison.

The difference in taxon abundance between healthy and tumorous tissue at the phylum level can be explained by the higher abundance of *Bacteroidetes* in healthy tissues and the higher abundance of *Proteobacteria* and *Actinobacteria* in tumor tissues. Our results are partly contradictory to the results of Parra-Grande et al. [30]; the taxon richness of the tumor tissue was found to be lower by both studies, but there is a difference between the observed *Actinobacteria* abundance values. While only males were enrolled in our negative control group, there were three times more male participants in the tumor groups as well. Clearly the gender of the participants alone cannot explain the reversal of the proportion of the *Actinobacteria* phylum. The previous study classified cancer patients into two clusters based on the most common genera, but no *Lachnospira* dominance (Cluster 1) was found among our samples. The *Staphylococcus* genus (Cluster 2) or the *Corynebacterium* genus were the most abundant in the samples we examined. Our tumor samples also contained *Propionibacteium* genus with a significantly higher abundance than healthy tissues. The latter two high-abundance genera, which belong to the *Actinobacteria*, can also explain the higher abundance of the phylum in our assay. The strains belonging to the genera *Staphylococcus*, *Propionibacterium*, and *Corynebacterium* found in high abundance in our bladder tumor samples play a role not only in the development of skin tumors but also in the development of gastrointestinal and urinary tract tumors [5,31,32,33,34,35,36]. The reason for the differences between the results of Parra-Grande et al. and ours—that we described different taxon abundances—can be that we used different methods to analyze the microbiome.

Bladder tumor samples from patients who received antibiotic therapy within 3 months showed a significantly different microbiome composition than healthy or antibiotic-free tumor samples. This can be partly explained by the fact that this group included individuals who were presumed to suffer from an acute bacterial infection. Bacterial species were present in their samples with a high abundance of between 34 and 97%. Although no bacteria were cultured from the urine of these eight patients, an acute infection of the mucous membrane could not be ruled out due to the dominant genus appearing with high abundance. The fact that potential mucosal pathogens were not cultured from the urine can be explained, on the one hand, by the fact that the urine and tissue microbiomes are different. On the other hand, there are genera that cannot be detected by routine aerobic culture. However, it was also confirmed that AB3 group members without acute infection had different microbiome compositions depending on the antibiotic used for the treatment.

Grouping the AB3 samples based on the three most frequently used antibiotic treatments, we experienced a significant difference in both the bacterial composition and the associated hBD levels. Among the differences between the healthy and tumor microbiome, a significantly higher abundance of *Staphylococcus* can be verified after treatment with all three antibiotics. Fibronectin-mediated attachment of BCG to the bladder mucosa is crucial for the development of an antitumor response [37], but this binding can be weakened competitively by Staphylococcal fibronectin-binding proteins [38]. The *Acinetobacter* genus, which was not detected on the healthy mucosa, was present with a significantly high abundance in the group treated with Fluoroquinolone. Assuming that Fluoroquinolone contributes to the selection of nosocomial multiresistant *Acinetobacter* strains, this therapy does not directly affect the antitumor treatment, but results in a decrease in prognosis time of survival. Although in vitro studies have demonstrated that Fluoroquinolones can be effective in the treatment of genitourinary tumors by inhibiting DNA metabolism of eukaryotic cells [39], in vivo microbiome changes should be considered when recommending therapy. In the antibiotic-treated groups, unlike the healthy mucosal microbiome, a significant excess of *Propionibacterium* and *Corynebacterium* genera was confirmed. *Propionibacterium* and *Corynebacterium* have been shown to increase PD-L1 expression on cancerous urothelial cells [40], and as a result, they can indirectly reduce the effectiveness of check-point inhibitor therapy.

The applied antibiotic treatments resulted in differently elevated hBD expressions. The most astonishing, 17-fold, increase in hBD3 expression occurred after the use of Beta-lactam antibiotics. Several previous studies have pointed out that hBD3 is overexpressed in head and neck tumors, and that it increases the proliferation of cells in cervical tumors [41,42]. Bacterial LPS is shown to increase hBD3 expression in oral squamous cell tumors, but this is untrue for colon carcinoma cells. In colon carcinoma, tumor-infiltrating monocytes are the reason for increased hBD3 levels, which in turn reduces the migration ability of the tumor cells [43]. Further research is needed to establish whether the increased hBD3 expression to different extents is a consequence of the microbiome change, or whether antibiotic therapies directly affect hBD3 production. The effect of elevated hBD3 levels on the progression of bladder carcinoma also requires considerable attention.

## 4. Materials and Methods

### 4.1. Sample Collection

Our recent microbiome-defensin molecular investigation compared 60 bladder cancer tissue samples in addition to 10 healthy bladder mucosal samples. From February to August 2021, those 60 BC patients were included in the study, who underwent tumor removal surgery at the MFUTH Urology Department and met inclusion and exclusion criteria. The inclusion criterion was the first surgical treatment of bladder carcinoma. Exclusion criteria were previous surgery, BCG treatment, previous antitumor chemotherapy or radiotherapy, and use of probiotics in the last 6 months. We selected 10 benign prostatic hyperplasia (BPH) patients of similar age who were operated on at the same time as the control group. Exclusion criteria were antibiotic or probiotic therapy within 6 months. A positive bacteriological culture of the urine voided before surgery was an exclusion criterion for both BC and BPH patients. The samples removed during transurethral resection (TUR) were divided for histological and microbiome-defensin expression studies. The tumor samples were classified into 3 groups: antibiotic treatment was not performed during the preceding half year, antibiotic treatment was performed more than 3 months before sample collection, and antibiotic treatment was performed within the last 3 months of sampling. The characteristics of the study participants are presented in Table 1.

### 4.2. DNA Isolation, 16S rRNA Gene Libary Preparation, and MiSeq Sequencing

From tissue samples after enzymatic dissolution with ProtK (56 °C, 5 h), DNA isolation was performed by ZymoBIOMICS DNA Miniprep Kit (Zymo Research Corp., Tustin, CA, USA). The V3–V4 region of bacterial 16S rRNA gene was amplified with tagged primers. PCR and DNA purifications were performed according to Illumina’s protocol. PCR product libraries were assessed using DNA 1000 Kit with Agilent 2100 Bioanalyzer (Agilent Technologies, Waldbronn, Germany). Illumina MiSeq platform (Illumina, San Diego, CA, USA) and MiSeq Reagent Kit v3 (600 cycles PE) were used to sequence the equimolar concentrations of pooled libraries.

Extraction negative controls and PCR negative controls were included in every run. All analysis procedures were conducted in duplicate from 2 separately isolated DNA samples of each patient. Raw sequencing data were retrieved from Illumina BaseSpace and data were analyzed by the CosmosID [44] bioinformatics platform (CosmosID, Germantown, MD, USA). The CosmosID-HUB Microbiome’s 16S workflow for taxonomy and species-level identification were conducted with DADA2’s naive Bayesian classifier, using the Silva version 138 database.

### 4.3. Defensin Expression Assays

After enzymatic dissolution with ProtK (56 °C, 5 h), total RNA from tissue samples was isolated by innuPREP RNA Mini Kit 2.0 (Analytik Jena GmbH, Jena, Germany) according to manufacturer’s instructions. Eighty to one hundred ng of RNA was used for RT-PCR assay performed using the PrimeScript RT reagent kit (Takara Bio, San Jose, CA, USA) and amplified the resulting cDNA on a qTOWER 3G (Analytik Jena GmbH, Germany) instrument in the presence of selected primers. The following primers were used for defensin expression assays: hBD1: 5′-TTG TCT GAG ATG GCC TCA GGT AAC-3′ forward, 5′-ATA CTT CAA AAG CAA TTT TCC TTT AT-3′ reverse, hBD2: 5′-CCAGCCATCAGCCATGAGGGTCTTG-3′ forward, 5′-CAT GTC GCA AGT CTC TGA TGA GGG AGG-3′ reverse, hBD3: 5′-AGC CTA GCA GCT ATG AGG ATC-3′ forward, and 5′-CTT CGG CAG CAT TTT CGG CCA-3′ reverse. The primers for GAPDH housekeeping gene were 5′-CTA CTG GCG CTG GCA AGG CTG T-3′ forward, and 5′-GCC ATG AGG TCC ACC ACC CTG CTG-3′ reverse.

Relative changes in mRNA expression were calculated by using the double delta Ct (ΔΔCt) method [45]. ΔCt of each tumor and BPH samples were normalized, calculating the difference between their gene of interest Ct value (hBD1, hBD2 or hBD3) and their Ct value of the GAPDH housekeeping gene. ΔΔCt values were calculated using the median ΔCt value of BPH samples considered as negative controls. The relative expression (RQ) fold change was calculated as 2^−ΔΔCt^.

### 4.4. Statistical Analysis

Levels of statistical significance (*p* < 0.05) for the difference between defensin expression rate and bacterial taxa abundances measured in the different cohorts was calculated by Mann–Whitney U test. Statistical significance between cohorts were implemented by Wilcoxon Rank Sum test for Chao1 alpha diversity and PERMANOVA analysis for Bray–Curtis Principal Coordinate Analysis (PCoA) Beta diversity using the statistical analysis support application of CosmosID [44].

### 4.5. Ethical Considerations

Sample collection protocols were approved by the Ethics Committee (EC) of Markhot Ferenc University Teaching Hospital (MFUTH) and EC of Semmelweis University (SE RKEB: 100/2018/100-1/2018/2021). The study was conducted in accordance with the Declaration of Helsinki ethical standards that promote and ensure respect and integrity for all human subjects. Patients treated at the MFUTH’s Urology Department between February and August of 2021 were enrolled in this study. All research was performed in accordance to guidelines and regulations of MFUTH. Written informed consent was obtained from all patients. All study participants gave written informed consent that data from their personal test results could be published. Data and test results in the manuscript cannot be linked to individual participants because all tests were anonymized.

## 5. Conclusions

Undoubtedly, antibiotic treatments used concomitant with various antitumor therapies worsen the prognosis for recovery. Here, we show that the microbiome composition of tumor tissue characteristic of an individual can change differently depending on the type of antibiotic therapy. Three months after the antibiotic treatment, no deviation from the microbiome of the untreated group was confirmed. The various antibiotic treatments affect not only the composition of the microbiome, but also the amount of human beta defensins produced in the tumor. An exceptionally high hBD3 expression was observed when beta-lactam therapy was used.

In the era of personalized medicine, it is crucial to understand the interactions between the microbiome and the amounts of human beta defensins appearing in its environment, as well as their combined effect on the tumor treatment options. With follow-up tests, it is necessary to reveal what microbiome compositions and what associated defensin levels are beneficial for different tumor treatment protocols.

Accurate knowledge of microbiome manipulation with antibiotics may even result in an increased response to tumor treatment in bladder cancer patients.

## Figures and Tables

**Figure 1 ijms-25-04562-f001:**
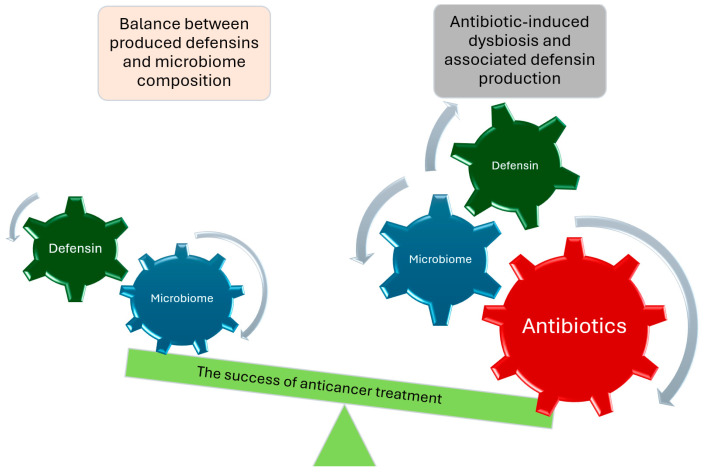
Presumed effect of antibiotic therapy during tumor treatment.

**Figure 2 ijms-25-04562-f002:**
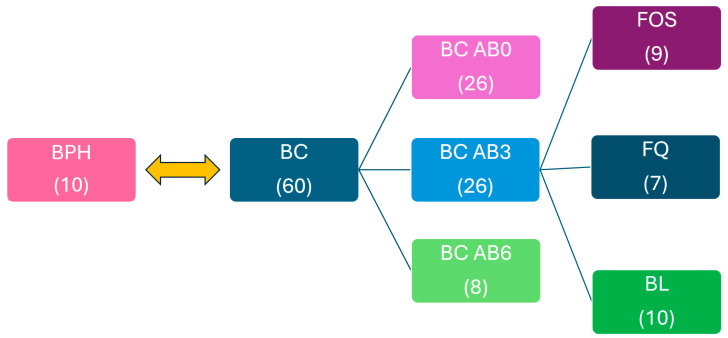
The study groups and the number of participants.

**Figure 3 ijms-25-04562-f003:**
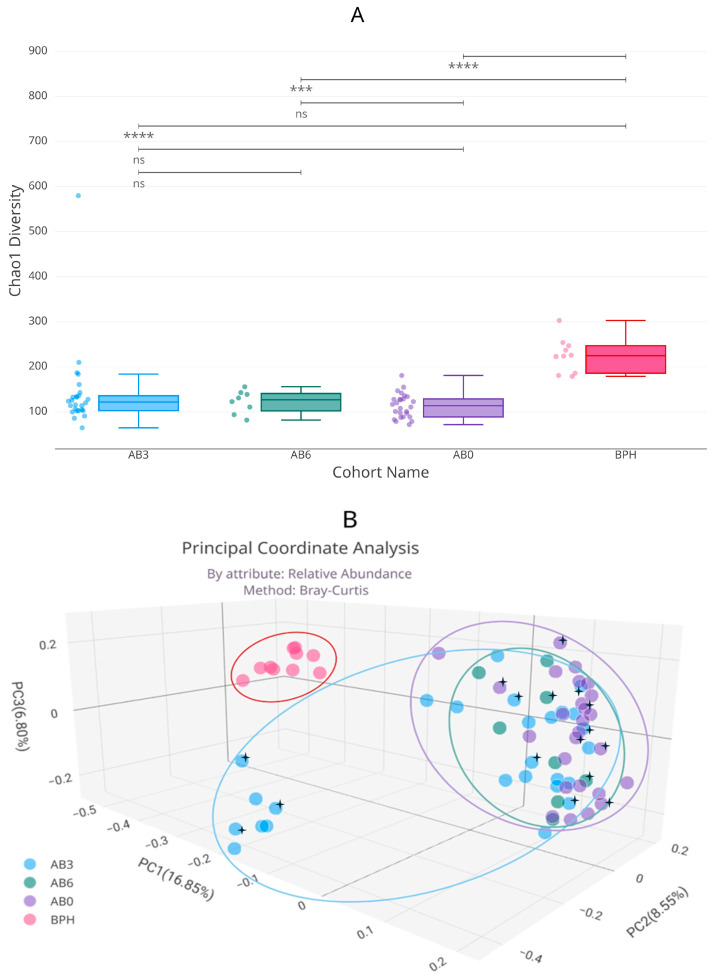
Chao1 alpha (**A**) and Bray–Curtis beta (**B**) diversity differences between BPH, AB3, AB6, and AB0 cohorts. The asterisk indicates female samples. ns: no significance, ***: *p* ≤ 0.001, ****: *p* ≤ 0.0001.

**Figure 4 ijms-25-04562-f004:**
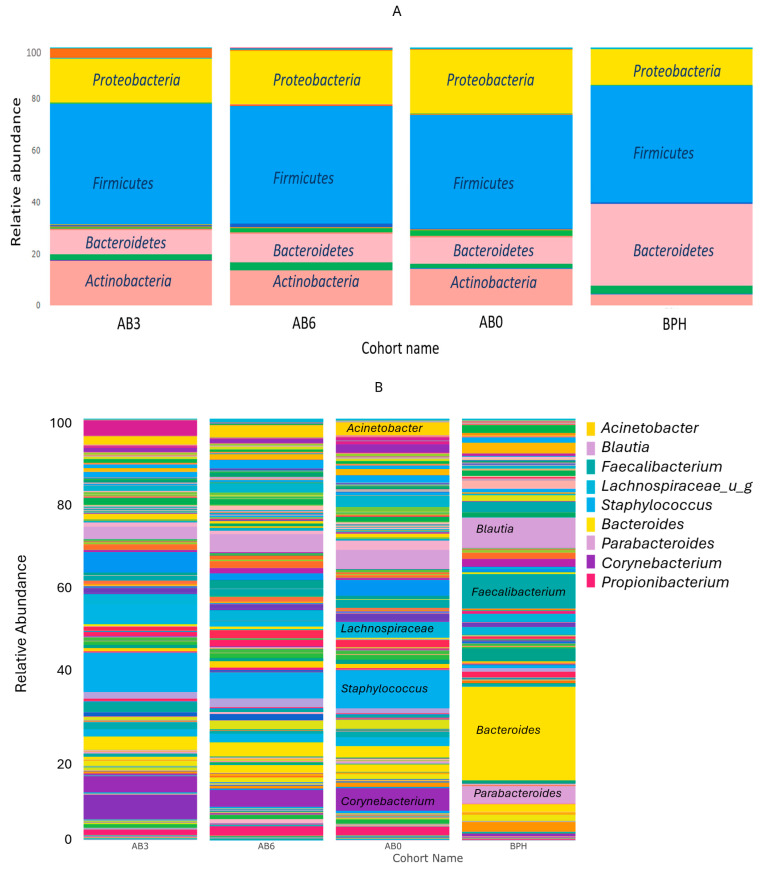
Taxon abundances at the phylum (**A**) and genus (**B**) level in AB3, AB6, AB0, and BPH cohorts.

**Figure 5 ijms-25-04562-f005:**
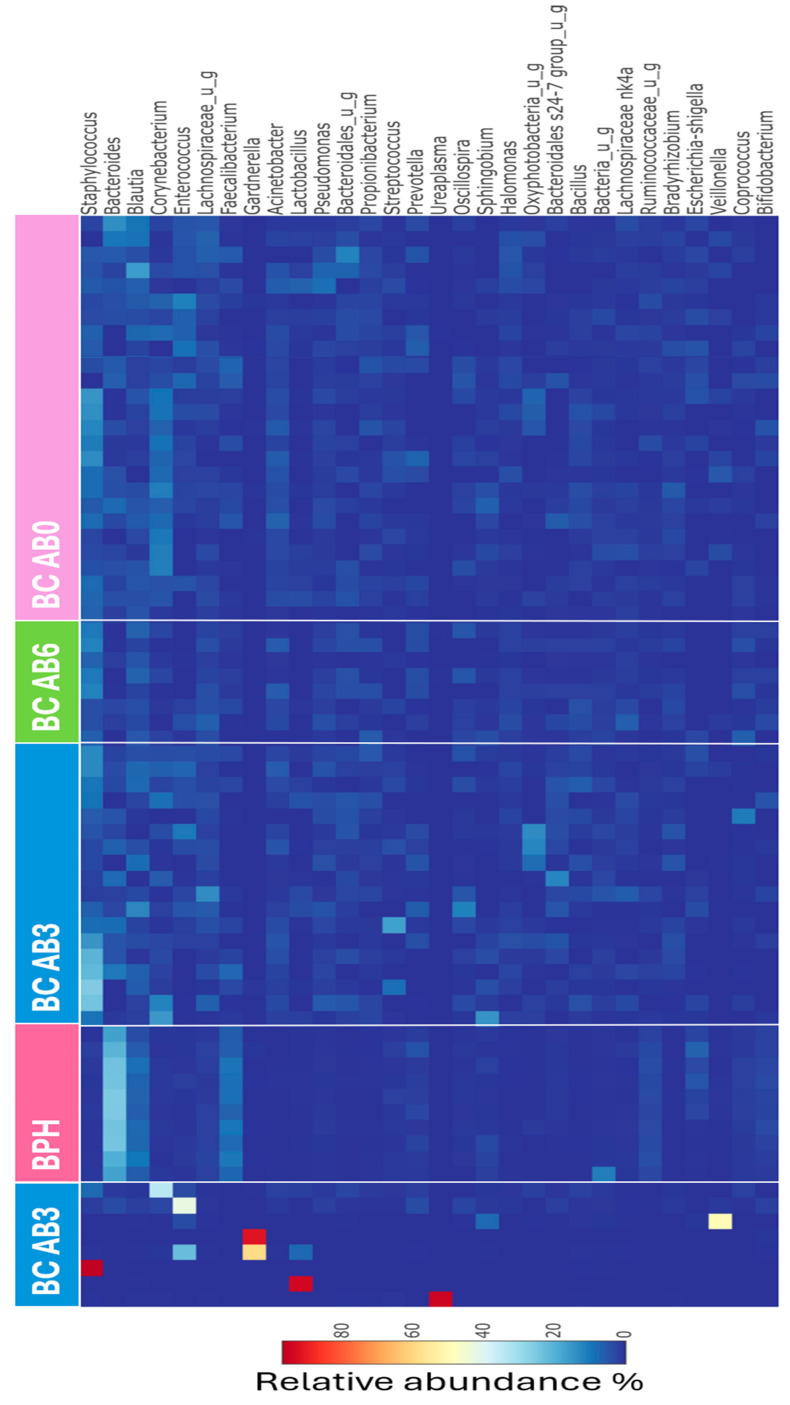
Heatmap visualization of the most abundant 30 genera in antibiotic-treated or untreated cancer tissues and in healthy mucosal samples. The thin columns represent the microbiome of patients; the relative abundance of genera increases from dark blue to dark brown shades.

**Figure 6 ijms-25-04562-f006:**
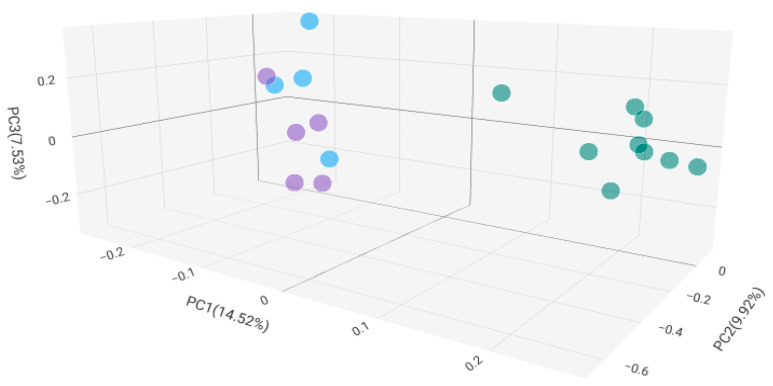
Bray–Curtis Beta diversity Principal Coordinate Analysis of antibiotic-treated patients’ samples. Purple: FOS; Blue: FQ; Green: BL.

**Figure 7 ijms-25-04562-f007:**
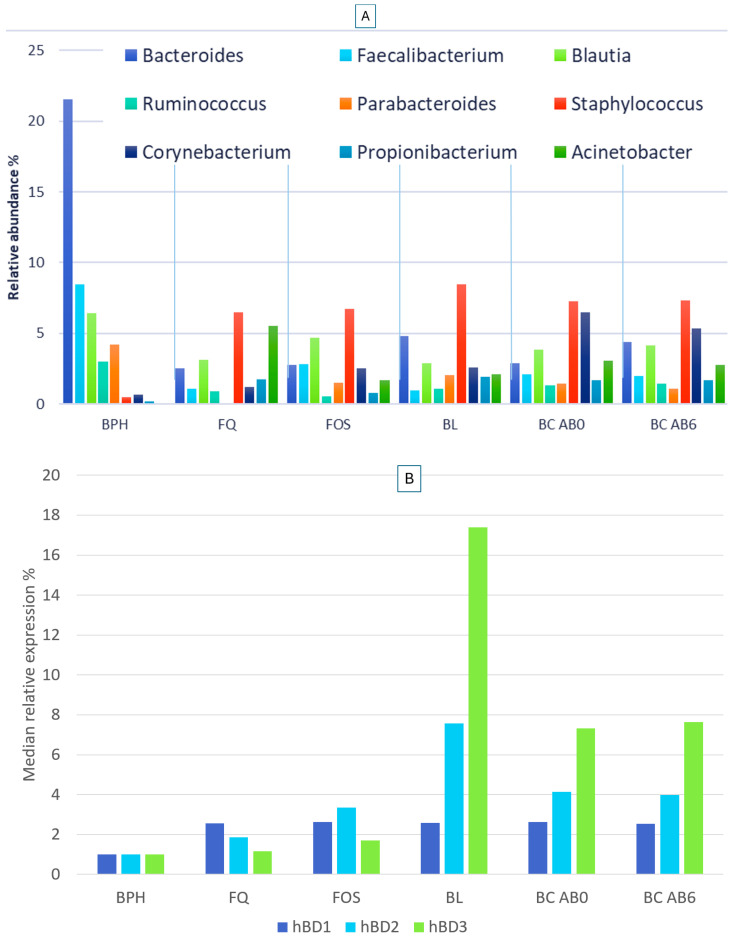
Relative abundance of bacteria at genus level (**A**) and related human beta defensin relative expressions (**B**) in patient tissues after pre-existing antibiotic treatments.

**Table 1 ijms-25-04562-t001:** Patients’ characteristics. BPH: benign prostatic hyperplasia; BC AB0: Bladder cancer without antibiotic therapy; BC AB3: Bladder cancer with antibiotic therapy in the last 3 months; BC AB6: Bladder cancer with antibiotic treatment more than 3 months before microbiome screening.

	Antibiotic Treatment Date	Number of Patients	Age Median (ICR)	Male/Female	Stage Ta/T1/T2/T3/T4	Grade G1/G2/G3
BPH	untreated	10	72 (17)	10/0	n.a.	n.a.
BC AB0	untreated	26	73 (15)	20/6	7/8/4/7/0	11/5/10
BC AB6	3 months<, 6 months>	8	74 (16)	6/2	2/3/1/2/0	3/3/2
BC AB3	3 months>	26	71 (19)	18/8	8/8/3/7/0	12/5/9

## Data Availability

The datasets generated and analyzed during the current study are available in the SRA repository: SRA/PRJNA 809202/www.ncbi.nlm.nih.gov (accessed on 22 February 2022).

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
