# Peer review of "Change in Tissue Microbiome and Related Human Beta Defensin Levels Induced by Antibiotic Use in Bladder Carcinoma"

_ijms, 2024, doi:10.3390/ijms25084562_

Round 1

Reviewer 1 Report

Comments and Suggestions for Authors

Monye et al., in the current manuscript, change in tissue microbiome composition and related human beta defensin levels caused by antibiotic use, possible factors influencing the treatment of bladder carcinoma. They observe microbiome composition of tumor tissue characteristic of an individual can be changed differently depending on the type of antibiotic therapy. Several reports have indicated that during antitumor therapy microbiome composition changes and sometime selective microbes can grow in this stress condition. A substantial revision and expansion are necessary before it can be considered suitable for publication in this journal.

1.      The author needs to show the clinical parameters with including and excluding parameters.

2.      Why does the author not consider gender to be a parameter? Because male and female have different physiology.

3.      As they said high hBD3 expression was observed when beta lactam therapy was used. Have they checked any serological parameters to prove this?

4.      Also to show the betta diversity of microbiome in the groups.

5.      Add specs after the value and symbols (like 10 %, in state 10%,),

6.      Additionally, include a schematic representation of the main theme of your work.

7.      Most of place species name not written in proper manner, make it uniform and as per guideline.   

8.      Write p-value in the state of p-value.

9.      The Discussion section requires expansion to offer a thorough interpretation of the findings, emphasizing their significance and potential implications in the context of antibiotics.

 prior to submission, the manuscript should undergo careful proofreading to ensure the proper usage and correctness of the English language throughout the document.

Comments on the Quality of English Language

NA

Reviewer 2 Report

Comments and Suggestions for Authors

The manuscript, entitled “Change in tissue microbiome composition and related human beta defensin levels caused by antibiotic use, possible factors influencing the treatment of bladder carcinoma,” explores whether, in addition to changes in the microbiome, changes in the related hBD levels can also cause the weakening effect of antibiotic therapy during tumor treatment. The target is of importance, and the findings of the present work highlight the potential area of research in the field. A few points will be necessary to accommodate in this study, which are:

  1. The title of the present research is purely classical, preferably reworded. The title should be creative, concisely composed, and relevant to the study’s objectives. In the abstract, the method and result parts are not clearly reflected according to the object, so the author should clarify them. Kindly re-check the mentioned key words for their suitability and rearrange them according to alphabetical grade.
  2. The introductory section has some gaps and lacks sound contemporary support from literature. Some lines of the introduction section need a thorough revision with the aid of appropriate literature. What is the need for mentioning lines 42–44? Please justify. The authors should elaborate on the impact of the sentences mentioned in lines 46–48. Please add some more relevant details in lines 54–55 to make it a more suitable statement. Further in line 55, what are some antibiotics? Explain. I would suggest authors smartly adjust the paragraph mentioned in lines 57–60 in their research objectives. Lines 63–65: How do authors ensure that no research group has so far published results on whether the composition of the microbiome detectable in bladder tumor tissue can be influenced by antibiotic treatment? Justify.
  3. In the Results section, small changes and modifications would make the manuscript catchier. This is my personal opinion: the structure of the manuscript, especially the addition of subheadings in Results, will be preferred; this should make the manuscript more structured and organized. In line 110, why authors used the word ‘valid bacterial sequences’, explain. Re-check the statement mentioned in 111–112. The inference obtained from principal coordinate analysis should be more elaborated. The title as well as the description of Figure 1 should be more descriptive, with all the required details, especially those shown in Figures 1a and 1b. Similarly, I found some incomplete details of figures 2(a and b) in the results; please reorganize and improve. What is the main purpose of mentioning Heatmap (Figure 3) in the present format? Explain. I think authors should re-visit Figure 3 and further segment this information into two distinct groups as a comparison (optional). In my opinion, line 185, where the use of the word ‘biggest’ is not appropriate, should be reworded. Why do figures 5a and 5b have two different resolution images? In the same figures, the quality of output should be more standardized (for the journal's level). Here, authors should check all figures and make the resolution uniform and better.
  4. The inferences drawn from the mentioned sections of results (including all figures and tables) were inadequately utilized in the discussion part. Please use this information and write a better discussion section. How you relate the findings (all) in the discussion should be applied (in the concerned section) and mentioned clearly. The strength of this work mainly relies on the well-written discussion section to significantly relate the obtained findings to established research. The discussion needs revision.
  5. The methodology is well written.

Minor comments

  1. I would suggest the authors add an appealing graphical abstract (workflow) with visuals, as shown in a variety of manuscripts, to enhance the understanding of their work.
  2. The research objectives section should be carefully mentioned at the end of the introduction part. 
  3. Table 1 should be restructured with a better interpretation status. Authors should consult some expert statisticians for better output analysis (generalized).
  4. Does this manuscript have any wider scope, or does it describe the solution to a specific problem? Justify the strengths and limitations of this work as to how it will benefit the readers. What is the main purpose for global readers to read this manuscript and utilize it in their research? Please explain.
  5. Add a conclusion section with future prospects to help readers retain the purpose of the article. Rewrite
  6. Some references (36, 37, and 44) are quite old; please check.

Comments on the Quality of English Language

 Minor editing of English language required

Reviewer 3 Report

Comments and Suggestions for Authors

Dear Authors,

thank you for providing this manuscript. Please find my comments per line and questions below:

The abstract is a little bit confusing to me. Could you please use some more lines to clarify the groups you tested and the outcome per group?

Line 62: As the microbiome is influenced by lots more than these two, please ad this here.

Line 62: Please give an explanation for the abbreviation FMT

Please explain what kind of antibiotic treatment you evaluated, as the different active ingredients could affect the microbiome differently.

Please add information about the subgroups in the Appendix Table 1.

Line 108: Please give an explanation for the abbreviation BPH at its first use.

Line 158: Please give an explanation for the abbreviation BL, FOS and FQ at its first use. (Table 1) Please introduce this subgroups with the number of patients in them.

Figure 5A/B: Could you please also add the results of the groups you introduced (AB0, AB6, AB3, BPH)? Did the hBD levels change?

Line 220: Did you evaluate the alpha diversity of the different antibiotic groups? Please add these data.

Line 238: Did you check gender associated differences in your study? Please add the data.

Question: Is it possible that the different abundancies between your study and the study of Parra-Grande et al. are caused by different techniques of sample collection, storage and extraction methods? Eventually a different region was amplified?

Line 272: please set in vitro to italic letters

Line 297: which parameters did you consider? Please add them.

Line 298: Please give an explanation for the abbreviation TUR.

Line 317: Could you please add the gene bank you used (Silver? GreenGenes?)?

Statistical analysis: The Mann-Whitney-U test and the Wilcoxon Rank Sum test are allowed for 2 groups only. Did you correct the results for the 4 groups you have in your study? Please use the correct test for more than two groups with the fitting post test, for example an one way or two way ANOVA with Bonferroni adjustment and a Dunn´s or Tukey post test.

Question: As far as other studies showed a gender difference I would encourage the Authors to recalculate their results for male patients only or try to balance the number of male and female patients.

kind regards

Comments on the Quality of English Language

Dear Authors,

The used English language is good to read. The Abstract is, maybe due to the brevity, not clear for me. Please use some more lines to clarify groups and outcome. Thank you.

kind regards

Reviewer 4 Report

Comments and Suggestions for Authors

The manuscript titled "Change in Tissue Microbiome Composition and Its Relation to Human Health" presents promising research. However, several revisions are necessary before recommending it for publication in IJMS.

Below are my comments:

  1. The title is excessively long and should be shortened for clarity and conciseness.
  2. The first two sentences of the abstract can be combined to avoid unnecessary repetition and conserve space.
  3. The final sentence in the abstract should be revised for clarity and significance.
  4. Regarding the statement "With our previous study, we proved that the microbiome...," it would be beneficial to provide more context on the authors' prior research to differentiate it from the current study.
  5. The statement regarding antibiotics and cancer therapy should be nuanced to reflect the complex balance between therapeutic benefits and potential drawbacks, particularly in the context of cancer treatment.
  6. The introduction should provide more detailed information about the methodology and objectives of the current study.
  7. Figure 2 requires improvements to the legend to enhance readability, possibly by enlarging the font size.
  8. Figure 3 needs correction and completion of descriptions to meet the standards for publication in IJMS.
  9. Consider reducing the size of signatures in Figure 5 for better visual presentation.
  10. The explanation of how Actinobacteria influence the abundance of the phylum should be clarified.
  11. The statement regarding the different microbiome compositions in the AB3 group members without acute infection requires further explanation and integration into the manuscript.
  12. The summary, particularly the second paragraph, should be expanded to provide a comprehensive overview of the findings and implications of the study.

In conclusion, while the manuscript presents valuable research, it requires significant revisions to meet the standards for publication. I recommend a major revision.

Round 2

Reviewer 3 Report

Comments and Suggestions for Authors

Dear Authors,

Thank you for taking my suggestions into account. I have only 2 small pojnts which I may ask you to correct:

Line 118/120 ff: please correct the bold letters.

Figure 5: Is it possible to turn (only) the name “Relative abundance %” up side down? Please check the bold letters.

kind regards

Author Response

Dear Reviewer,

We are grateful for your further improvement suggestions, we corrected the errors. Bold letters were found in lines 120-131, which were unjustifiably in this format, we made the requested correction here.

We turned the titel "Relative abundance %" up side down on Figure 5, and corrected the bold letters in the figure legend.

Sincerelly yours

Eszter Ostorházi

Reviewer 4 Report

Comments and Suggestions for Authors

The authors have greatly improved this article. I believe it is now entirely suitable for publication in IJMS.

Author Response

Dear Reviewer,

We thank you again for your help, which helped our manuscript to improve. We are grateful for your kind support for the publication of our manuscript in IJMS.

Sincerelly yours

Eszter Ostorházi